# Effects of La Addition on Microstructure Evolution and Thermal Stability of Cu-2.35Ni-0.59Si Sheet

**DOI:** 10.3390/ma16114105

**Published:** 2023-05-31

**Authors:** Mingfei Wang, Shuaifeng Chen, Songwei Wang, Mengxiao Zhang, Hongwu Song, Shihong Zhang

**Affiliations:** 1School of Rare Earths, University of Science and Technology of China, 96 Jinzhai Road, Hefei 230026, China; wangmf123@mail.ustc.edu.cn; 2Ganjiang Innovation Academy, Chinese Academy of Sciences, 1 Science Academy Road, Ganzhou 341119, China; 3Shi Changxu Advanced Materials Innovation Center, Institute of Metal Research, Chinese Academy of Sciences, 72 Wenhua Road, Shenyang 110016, China; chensf@imr.ac.cn (S.C.); swwang16b@imr.ac.cn (S.W.); mxzhang@imr.ac.cn (M.Z.); hwsong@imr.ac.cn (H.S.)

**Keywords:** Cu-Ni-Si, Cu-2.35Ni-0.59Si alloy, La-rich phase, microstructure evolution, precipitate behavior, thermal stability

## Abstract

A Cu-2.35Ni-0.69Si alloy with low La content was designed in order to study the role of La addition on microstructure evolution and comprehensive properties. The results indicate that the La element demonstrates a superior ability to combine with Ni and Si elements, via the formation of La-rich primary phases. Owing to existing La-rich primary phases, restricted grain growth was observed, due to the pinning effect during solid solution treatment. It was found that the activation energy of the Ni_2_Si phase precipitation decreased with the addition of La. Interestingly, the aggregation and distribution of the Ni_2_Si phase, around the La-rich phase, was observed during the aging process, owing to the attraction of Ni and Si atoms by the La-rich phase during the solid solution. Moreover, the mechanical and conductivity properties of aged alloy sheets suggest that the addition of the La element showed a slight reducing effect on the hardness and electrical conductivity. The decrease in hardness was due to the weakened dispersion and strengthening effect of the Ni_2_Si phase, while the decrease in electrical conductivity was due to the enhanced scattering of electrons by grain boundaries, caused by grain refinement. More notably, excellent thermal stabilities, including better softening resistance ability and microstructural stability, were detected for the low-La-alloyed Cu-Ni-Si sheet, owing to the delayed recrystallization and restricted grain growth caused by the La-rich phases.

## 1. Introduction

With the arrival of the information age, integrated circuits and chips are quickly developing with a wider range of application [1,2]. In integrated circuits and chips, the lead frame is the key component and it is usually made of copper alloys with good mechanical properties, electrical properties and thermal stability [2]. In these copper alloys, Cu-Ni-Si alloys are a commonly used lead frame material, with a tensile strength of about 700 MPa and a conductivity of around 45 %IACS (International Annealed Copper Standard) [2,3,4,5]. As a typical precipitation-strengthening alloy, the mechanical properties of Cu-Ni-Si alloys are mainly controlled by precipitates of Ni_2_Si and Ni_3_Si [4]. Additionally, the conductivity in Cu-Ni-Si alloys is also closely related to precipitate behavior [5,6]. Therefore, the reasonable and effective control of microstructure evolution and precipitation behavior are of great importance for the attainment of comprehensive strength and conductivity properties.

In general, there are two main methods for improving the comprehensive properties of Cu-Ni-Si alloys. The first is the modification of the preparation process by adjusting the combination of plastic deformation and heat treatment [5,6,7,8]. Adjusting process parameters to improve alloy properties is usually used in aging-strengthened alloys. The most important process steps in this type of alloy are solid solution and aging processes, and adjusting the process parameters is a more convenient approach. Aging-strengthened alloys are not only available in Cu-Ni-Si alloys, but also in high-strength and high-elastic copper alloys such as Cu-Ni-Sn. Aging-strengthened alloys undergo amplitude-modulated decomposition and discontinuous precipitation during the aging process, and the precipitates undergo a specific transformation during the aging process [3,8]. Compared with adjusting the preparation process, adding alloy elements to Cu-Ni-Si alloys is considered a more effective means of adjusting the microstructure and precipitation behavior of the alloy. Common additive elements include Al, Mg, Zr, Cr, Ti and rare earth elements [9]. The addition of Al refines the casting microstructure of Cu-Ni-Si-Al alloys, which promotes the precipitation, and effectively enhances the anti-stress relaxation property [10,11]. With Mg, Cu-Ni-Si-Mg shows higher strength as a result of decreasing the inter-precipitate spacing, as well as an improved anti-stress relaxation capability [12]. The addition of Zr can remove impurities and improve the purity of the matrix, but it destroys the precipitation-strengthening effect of subsequent precipitates and decreases the strength of the alloy [5,13]. Alloying with Cr can significantly reduce the grain size and enhance the mechanical properties without deteriorating the electrical conductivity [5,14]. The content of Ti affects the precipitation behavior of precipitates during aging, resulting in changes to the properties of the alloy [15,16,17]. 

In recent years, more attention has been paid to rare earth elements because of their special physical and chemical properties, which can effectively modify the microstructures and the related properties of alloys [18,19]. As light rare earth elements, La and Ce have similar physical and chemical properties, and both have the characteristics of low cost and abundance. In addition to its known function of purifying melt and refining grain, research on La and Ce as alloying elements is also gradually increasing. It has been reported that the addition of Ce can increase the mechanical strength and refine the size of precipitates [20,21,22,23,24]. For Cu-Ni-Si alloys, Ce can improve the recrystallization temperature and affect the precipitation behavior, which results in a good match between strength and electric conductivity [25]. Chen et al. [26] suggested that a small amount of La retards the recrystallization of pure copper, while a large amount of La accelerates the recrystallization. Additionally, some studies also show that the joint addition of two or more alloy elements can improve the alloy properties [9,27,28]. However, there is still limited research clarifying the effect of rare earth elements on microstructure evolution, formation of precipitates and comprehensive properties in Cu-Ni-Si alloys, especially the interaction mechanisms between La additions and precipitate kinetics during overall processing.

In this paper, the effects of La, added as alloying element, on the microstructure evolution and precipitation behavior of a Cu-2.35Ni-0.59Si alloy sheet, were systemically studied. The change in comprehensive properties and thermal stability, with the addition of La, were compared. The effects of the mechanism of the La-rich phase, during the sheet preparation process, were clarified.

## 2. Experimental Procedures

### 2.1. Material Preparation

The investigated material in this study was Cu-2.35Ni-0.59Si-0.11Mg-0.069Zn-XLa (in wt.%), with X being 0, and 0.05, respectively. The raw materials were electrolytic copper (99.99 wt.%), electrolytic nickel (99.96 wt.%), industrial silicon (98 wt.%), high purity magnesium ingot (99.9 wt.%), high purity zinc ingot (99.995 wt.%) and pure lanthanum (99.5 wt.%) (all raw materials come from Shenyang Jiabei Trading Co., Ltd., (Shenyang, China). Alloy ingots were prepared using a vacuum induction melting furnace. The specific compositions of Cu-2.35Ni-0.59Si alloys were measured with ICP (Inductively coupled plasma emission spectrometer, Thermo Corporation, Waltham, MA, USA), as listed in Table 1.

Figure 1 displays the entire preparation process of the studied alloy. After casting, the ingots were homogenized at 920 ℃ for 2 h in a resistance furnace, and hot-forged to a thickness of 30 mm followed by air-cooling. The purpose was to eliminate dendrite and segregation during casting, as well as to generate improved rolling ability to avoid cracking. Then, the forged billets were heat treated at 900 ℃ for 2 h, and hot rolled to a thickness of 6 mm, with reduction ratio of 80% via multiple passes. Subsequently, solid solution at 820 ℃ for 1 h was performed. The solution sheet was then cold rolled to a thickness of 1 mm, and aged at 450 ℃ for varied hours, followed by water quenching.

### 2.2. Microstructure and Property Characterization

The surface observed during the experiment was the entire RD-TD surface, that is, the contact surface between the alloy plate and the roll. The sample of a corresponding size was taken from the center of the alloy plate, which was ground and polished at the RD-TD surface, then etched with a mixture reagent of FeCl_3_ (5 g), HCl (5 mL) and C_2_H_5_OH (100 mL). The microstructures of the samples were observed via an optical metallographic microscope (OM, Axio Observer Z1, Zeiss, Oberkochen, Germany). The second phases in different state samples were observed using a scanning electron microscope (SEM, Sigma-500, Zeiss, Oberkochen, Germany) with energy dispersive spectroscopy (EDS). A transmission electron microscope (TEM, TecnaiF20, FEI Company, Hillsboro, OR, USA) was employed to further examine the second phase and crystal structure. The precipitate kinetic of phases was determined by differential scanning calorimetry (DSC, STA449F3, NETZSCH, Selb, Germany) in an Argon atmosphere, from 300 °C to 1150 °C, at constant heating rates of 5 °C/min, 10 °C/min, 20 °C/min and 30 °C/min, respectively. The OM and TEM images were obtained statistically by Image Pro Plus 6.0 software (Software version: 6.0, Media Cybernetics, Rockville, MD, USA) to obtain the size of the grains and the precipitates [29]. 

The hardness measurement was carried out on a Vickers microhardness tester (LM247AT, LECO, Saint Joseph, MI, USA), with a measurement error of ± 5% at no less than 5 positions per sample. The electrical conductivity was obtained by measuring its resistivity (LSR Seebeck, Linseis, Zehlb, Bavaria, Germany) and utilizing the conversion relationship between resistivity and conductivity. Due to the need for alloys to serve at high temperatures, it is crucial to evaluate the thermal stability of materials [30]. The high-temperature stability comparison experiment involves selecting an alloy sheet after cold rolling, and conducting a heat treatment at 850 °C for 3 h, followed by observing the microstructure at room temperature.

## 3. Results

### 3.1. Microstructure Features after Hot Rolling and Solid Solution Treatment

Figure 2 shows the microstructure (OM) of hot-rolled and solid solution Cu-2.35Ni-0.59Si alloys. In Figure 2a to Figure 2d, fully recrystallized microstructures, obtained for two hot-rolled alloys, are shown, mostly within in Figure 2. The grains of the hot-rolled alloy were equiaxed (Figure 2a,c). With the addition of 0.05 wt.% La, the grains became smaller with a reduced average size of 19.8 μm, down from 23.5 μm in La-free alloys. La, which causes the alloy to produce fine, La-rich phases, was determined to be mainly distributed at the grain boundary. In addition, the fractions of the La-rich phase in the hot-rolled alloys are 0% and 0.37%.

After subsequent solid solution treatment (Figure 2b,d), a distinctive microstructure evolution was discerned within two alloys. For the La-free sheet (0 La in Figure 2b), solution-treated grains were obviously coarsened, with a statistical size of about 118.7 μm. In particular, there were large-sized lamellae annealing twins inside the matrix (marked by red circle). As for 0.05 wt.% La-alloyed samples, their microstructures were much refined, with small lamella twins, and an average grain size of 31.2 μm. That is, the grain size when no La was added (0 wt.% La) was 3.8 times larger than that with a low amount of La (0.05 wt.% La). Moreover, La-rich particles in 0.05 wt.% alloy samples were mainly observed at the twinning or grain boundaries. According to statistics, the fractions of the La-rich phase in the solid solution alloys are 0% and 0.32%. Such differences in microstructure evolution indicate distinctive recrystallization and grain growth behaviors for Cu-2.35Ni-0.59Si alloys with a La addition, which is likely to be closely related to the variations in distribution and size of the La-rich phase. A detailed explanation is in the following Discussion section.

Figure 3 shows the SEM images of hot-rolled and solid solution Cu-2.35Ni-0.59Si alloys. With a La addition of 0.05 wt.%, the La-rich phase appears in hot-rolled sheets with an island shape (Figure 3c). The element proportion of the La-rich phase is Cu_24.7_La_14.8_Ni_25.5_Si_35_ (at%), with an average size of about 3 μm. After subsequent solid solution treatment, the second phases with white, bright, spherical and distributed, at the grain boundary were discerned, as shown in (Figure 3b,d). For the La-free sheet (0 La in Figure 3b), the new second phase, with a size of about 800 nm, appeared. These second phases accounted for 3.8% in alloy, and their EDS results are Cu_49.2_Ni_30.7_Si_20.1_ (at%). The EDS results show that the phase atomic ratio of Ni and Si is 3:2, and this was speculated to be the Ni_3_Si_2_ phase [31]. With regard to the 0.05 wt.% La sample, the content of Ni_3_Si_2_ was 2.3%. That is, with the addition of La content, the fraction of the Ni_3_Si_2_ phase decreased. The proportion of Ni_3_Si_2_ was different, due to the La element, combined with the Ni and Si atoms, to form a La-rich phase.

The composition and element proportion of the La-rich phase in the alloy were almost unchanged after the solution treatment, which showed that the composition of the La-rich phase remained stable during the heat treatment process (Figure 3). This can be explained by the difference in the electronegativity of elements. Compared with Cu, La is easier to combine with Ni and Si atoms to form a La-rich phase, the reason for this being the observation difference in electronegativity values. The electronegativity values of La, Ni, Si and Cu are 1.11, 1.92, 1.98 and 1.90, respectively [32]. Therefore, the subsequent aggregation and distribution of the Ni_2_Si phase cannot have been caused by the precipitation and combination of Ni and Si atoms in the La-rich phase. From the EDS results obtained from the line scanning of the La-rich phase in the alloy, it can be deduced that the solid solution treatment caused the aggregation distribution of Ni and Si atoms around the La-rich phase. The width of the Ni and Si atom accumulation region, around the La-rich phase in the 0.05 La alloy, was about 0.7 μm. (Figure 4).

### 3.2. Second Phase in Different La-Content Alloys after Aging Treatment

Figure 5 displays the SEM images of the second phase (including primary and precipitate phases) of different La-alloyed sheets after aging treatment. As shown in Figure 5a, a large number of spherical Ni_3_Si_2_ phases were dispersed in 0 La alloy, with an average size of about 500 nm. The specific element contents of Ni_3_Si_2_ were measured to be Cu_45.2_Ni_33.9_Si_20.9_ (at%), which was similar to the solid solution state. From the local magnified regions of A and B, the Ni_2_Si phases in the La-free alloy exhibited fine sizes and uniform distribution. With the addition of 0.05 wt.% La (Figure 5b), the number of Ni_3_Si_2_ phases decreased, and the La-rich phase showed dispersed distribution. The La-rich phase, with contents of Cu_15.7_La_6.8_Ni_46.9_Si_30.6_ (at%), exhibited less of a difference from the solid solution state, which suggests that there was good thermal stability of the La-rich phase. Meanwhile, it was found that the La-rich phase had a considerable influence on the precipitate behavior during aging. Known from enlarged regions, the density of the Ni_2_Si phase around the La-rich phase was much higher (region D) than that in region C, indicating that the ununiform distribution of the Ni_2_Si phase was a result of the La addition. Theoretically, the precipitate of Ni_2_Si particles is controlled by a diffusion process, via spinodal decomposition [31,33]. The evolving distribution of the Ni_2_Si phase with different La additions, as described above, is mainly caused by the different energy storage around the La-rich phase.

Figure 6 shows TEM images of aged alloys with different La contents, in which the Ni_3_Si_2_ and Ni_2_Si are clearly distinguished in size and contrast. In the 0 La aged alloy, Ni_3_Si_2_ phases were observed to be spherical, with a size of 200~300 nm (Figure 6a). Meanwhile, spherical Ni_2_Si phases had a size of 50~80 nm [31,33,34,35]. With the addition of 0.05 wt.% La, the island-like La-rich phase presented as an irregular shape, and the average size exceeded 2 μm. Compared with the 0 La alloy, the number of the Ni_3_Si_2_ phase decreased significantly, while the density of the Ni_2_Si phase around the island-like La-rich phase significantly increased (Figure 6c,d). According to the statistical analysis of the size of the Ni_2_Si phase in the TEM pictures, the average sizes of the Ni_2_Si phase in the two alloys were 27.5 nm and 30.9 nm, respectively. With the addition of La, the uniformly distributed Ni_2_Si phase in the matrix disappeared.

The selected area electron diffraction (SAED) pattern in Figure 6b confirms the intermetallic phase Ni_2_Si. Ni_2_Si is a common precipitate in Cu-Ni-Si alloys; the crystal structure of Ni_2_Si is orthorhombic (the same as PbC12) and has lattice parameters a = 3.75 Å, b = 5.0 Å and c = 7.04 Å [36]. Similar to the precipitation phase, Ni_3_Sn is formed by spinodal decomposition in the Cu-Ni-Sn system; the discontinuous precipitation Ni_2_Si in Cu-Ni-Si alloy is also formed through spinodal decomposition [8,36,37,38]. In addition, both types of alloy undergo corresponding transformation processes during the formation of precipitation phases. The current research results show that when the aging temperature is between 400–450 °C, the phase transformation sequence of Cu-Ni-Si alloys is approximately as follows: super saturated solid solution→spinodal decomposition→ordering→Ni_2_Si precipitation [36,38].

### 3.3. Comprehensive Properties of Different La-Content Alloys after Aging Treatment

Figure 7 displays the hardness and electrical conductivity variation curves of alloys with different La content after aging treatments. As seen in Figure 7a, the hardness of the 0 La alloys increased with the prolongation of holding time, reaching a peak at 6 h, and the subsequent hardness remained stable. However, the peak hardness of the 0.05 La alloy appeared at 4 h, and subsequently demonstrated a decreasing trend in the prolongation of aging time. Moreover, Figure 7b shows the variation curve of electrical conductivity of the 0 La and 0.05 La alloys with aging time. Additionally, the electrical conductivity of both alloys increased with increasing aging time. The peak hardness of the 0 La alloy, during aging, was 243.5 HV, while for the 0.05 La alloy, it was 241.1 HV. The electrical conductivity of the 0 La alloy and the 0.05 La alloy could reach 45.7% IACS and 44.1% IACS, respectively, after aging for 12 h.

Based on the hindrance effect of the La-rich phase on grain boundary migration during heat treatment (Figure 2), it was speculated that it was beneficial for the improvement of the thermal stability of the alloy. Figure 8 shows the microstructure of the 0 La and 0.05 La alloys after cold rolling and holding at 850 °C for 3 h. The grain size of the 0 La alloy (Figure 8a) underwent severe coarsening after high-temperature and long-term heat treatment, while the grain size of low-content La alloy (Figure 8b) was much smaller than that of the La-free alloy. Further magnified by 500 times (Figure 8c,d), it can be observed that there are spherical or ellipsoidal La-rich phases, with a size of approximately 3.7 μm, on the grain boundaries and twin boundaries of the 0.05 La alloy. It is speculated that the improvement of thermal stability in the 0.05 La alloy is related to the pinning dislocation and the Zener drag effect of the La-rich phase.

## 4. Discussion

### 4.1. Correlation between La Addition and Microstructure Evolution

Figure 9 provides the mechanism for the strong correlation between La addition and microstructure evolution during hot rolling and solid solution. In Cu-Ni-Si alloys, the hindering effect increased, due to the increased proportion of the second phase, which can cause pinning at the grain boundary [31,33,34,35,39,40]. It has been reported that La and some other elements have the effect of refining the grain size in copper alloys [5,20,40,41,42,43,44]. Therefore, the Zener drag effect, caused by the La-rich phase, can pin the grain boundaries and hinder the grain growth, contributing the microstructure refinement [45,46,47,48]. That is, the La addition in Cu-2.35Ni-0.59Si alloys can induce the Zener drag effects by the formation of the La-rich phase, which can thus greatly affect the grain growth during hot rolling and solid solution processes.

During solid solution treatment, the growth of recrystallized grain and annealing twins mainly occurred on the basis of the fully recrystallized microstructure after hot rolling. As seen in Figure 2, the growth process of grains and twins was performed well in the La-free alloy. As a result, the grain size and twin width were much larger. For La-containing alloys, the growth behavior can be constrained, as a result of the Zener drag effect induced by La-rich phases. For the second phase with smaller size, spherical shape and more uniform distribution, a higher hindrance to the movement of the grain boundary can be induced, and the final grain size can become smaller [45,49]. Theoretically, the Zener drag pressure (*P_Z_*), generated by randomly distributed phases, can be expressed as [49]:(1)$$P_{Z}=\frac{3F_{v}\gamma_{b}}{2r}$$
where *γ_b_* is interfacial energy, *F_v_* is the particle fraction and *r* is the particle size. When *r* is smaller and *F_v_* is higher, the *P_Z_* is greater. The grain growth of the alloy ceased as the driving pressure (*P*) became equal to the pinning pressure (*P_Z_*) [45,46]. The proportions of La-rich phases (*F_v_*) in the 0 La and 0.05 La alloys were 0% and 0.32%, respectively, with average sizes of 0 μm and 1.68 μm, respectively. However, the large-sized La-rich phase was bypassed during grain boundary migration and was finally present in the grain. Most of the La-rich phases in the 0.05 La alloy were smaller than 1.5 μm (0.28%), which can effectively hinder the grain boundary migration.

### 4.2. Effects of La Element on Formation Kinetics and Distribution Characteristics of the Ni_2_Si Phase

In order to quantify the effect of the La-rich phase on the precipitation of the Ni_2_Si phase, DSC tests were used to calculate the activation energy of the precipitation reaction [50,51]. Figure 10 shows the DSC test curve, combined with theoretical Equation (2) [52,53] to calculate the activation energy with different La content (Figure 10c).
(2)$$\ln\left(\frac{b}{T_{p}^{2}}\right)=-\frac{E_{a}}{RT_{p}}+C$$
where *E_a_* is the activation energy of the precipitation reaction, *b* is the heating rate, *T_p_* is the peak temperature in the DSC curve. The exothermic peak strength of the precipitated phase near 450 °C increases with the increase of heating rate (red diamond in Figure 10). *R* is the ideal gas constant. *C* is a constant.

Taking ln(*b*/(*T_p_^2^*)) and −1/(*RT_p_*) as the X-axis and the Y-axis, respectively, linear relation can be established by substituting the parameters obtained from the experiments into Equation (2), as shown in Figure 10c The slope of the fitting line (*E_a_*) is the exact value of precipitation activation energy. The *E_a_* values of the 0 and 0.05 alloys were calculated as 36.2 kJ/mol and 33.0 kJ/mol, respectively. That is, the activation energy of the precipitate reaction of the Ni_2_Si phase decreases with the addition of La content. The reason for this decreased *E_a_* is closely related to the enrichment of Ni and Si elements, at the vicinity of the La-rich phase, after solid solution treatment (see Figure 4). With the decreased *E_a_* value after La addition, a high precipitation tendency of the Ni_2_Si phase can be anticipated for La-containing alloys. In addition, the melting point of the La-rich phase can be obtained through DSC experiments at approximately 957.75 °C.

The obvious difference in Ni_2_Si phase distribution, among two alloys displayed in Figure 5, proves that the La-rich phase had a prominent influence on the precipitate behavior of the precipitate phases. Figure 11 demonstrates the mechanism for the compact correlation of the precipitate behaviors between the La-rich phase and the Ni_2_Si phase’s precipitation behavior, which incorporates the influence of La-rich phase proportion. For alloys containing La, abundant dislocations aggregate around the La-rich phase during cold rolling, due to the impeding effect of the second phases. The distortion region and the large number of dislocations around the La-rich phases, induced during cold rolling, provided the driving energy for the precipitation of the Ni_2_Si phase during the subsequent aging treatment. Thus, the high density of the Ni_2_Si phase, around the La-rich phase, was mainly introduced by the combined effect of aggregation of Ni/Si atoms and high deformation energy. It was noted that the formation of the La-rich phase incorporated Ni and Si atoms. Thus, the proportion of the Ni_2_Si phase gradually decreased with the increase of La content [47,54]. This was deduced because of the difficulty of despoiling the Ni/Si atoms from the La-rich phase during aging, which was indirectly suggested by the Ni/Si content of the La-rich phase during full processing (Figure 3 and Figure 6).

### 4.3. The Effects of the La Element on The Properties of Aged Alloy Sheets

The addition of the La element had the following effects on the properties of the aged-state alloy (as shown in Figure 7). Firstly, the hardness of the alloys differed, the dispersed distribution of the La-rich phases in the 0.05 La alloys resulted in grain refinement, due to the pinning effect of La-rich phases on grain boundaries, thereby improving the hardness. However, at the same time, the Ni and Si atoms, consumed by the formation of the La-rich phase, caused a decrease in the density of the Ni_2_Si phase in the matrix. Additionally, the characteristics of the Ni_2_Si phase aggregation and distribution, around the La-rich phase, resulted in a decrease in the dispersion-strengthening effect, which led to a decrease in hardness (as shown in Figure 11). Ultimately, the combined effect of the two aspects showed a slight decrease in the hardness of the 0.05 La alloy. The difference in alloy electrical conductivity is also closely related to the La element. Firstly, the La-rich phase reduced the activation energy of the Ni_2_Si phase precipitation (as shown in Figure 10), and promotion of the precipitation of the Ni_2_Si phase increased the electrical conductivity. However, the grain refinement, caused by the La-rich phases, led to an increase in grain boundary density, resulting in an increase in electron scattering probability and a decrease in electrical conductivity. Under the action of two mechanisms, the electrical conductivity of La-containing alloys was ultimately slightly lower than that of 0 La alloys. However, it was precisely because of the pinning effect of the La-rich phase on grain boundaries that the 0.05 La alloy had good thermal stability.

As shown in Figure 8, the grain size of alloys, after high-temperature treatment, had distinctive differences. The grain coarsening of the 0 La alloy was due to the absence of the second phase with good thermal stability. Therefore, during the high-temperature and long-term heat treatment process, a large number of dislocations, caused by cold rolling, moved quickly, leading to recrystallization. During the subsequent holding time, the recrystallized grains continued to grow. However, when La was added, the La-rich phases were formed in the alloy, and dislocations formed during cold rolling were aggregated and distributed around the La-rich phase, due to its pinning effect. Under high temperature conditions, the high-density dislocation zone around the La-rich phase promoted the rapid occurrence of recrystallization nucleation [45,49]. As the insulation time was extended, the La-rich phase with good thermal stability hindered the migration of grain boundaries, resulting in good thermal stability of the 0.05 La alloy sheet.

## 5. Conclusions

In this study, Cu-2.35Ni-0.59Si alloys with a La addition were designed, and the microstructure of the alloys in the preparation process was studied. The comprehensive properties of Cu-2.35Ni-0.59Si alloy sheets, and alloys with containing La, were compared. The following conclusions have been drawn:(1)The pinning effect of the La-rich phase, on grain boundaries in alloys, results in grain size refinement.(2)The addition of the La element reduces the precipitation activation energy of the alloy, resulting in the aggregation and distribution of the Ni_2_Si phase around the La-rich phase.(3)The prepared alloy has excellent comprehensive properties, with a hardness of 239.8 HV and an electrical conductivity of 44.1% IACS for the 0.05 La alloy.(4)The La-rich phase has good thermal stability, and its pinning effect on grain boundaries significantly improves the thermal stability of 0.05 La alloy sheets.

## Figures and Tables

**Figure 1 materials-16-04105-f001:**
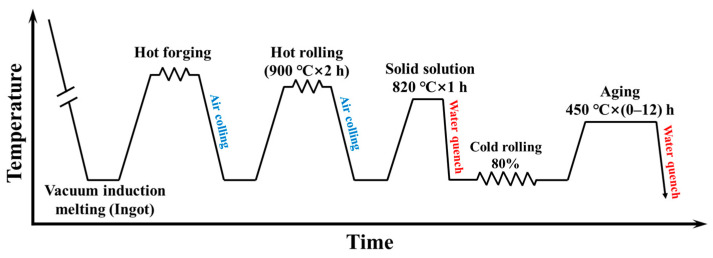
Flowchart of the entire preparation process of studied alloy sheet.

**Figure 2 materials-16-04105-f002:**
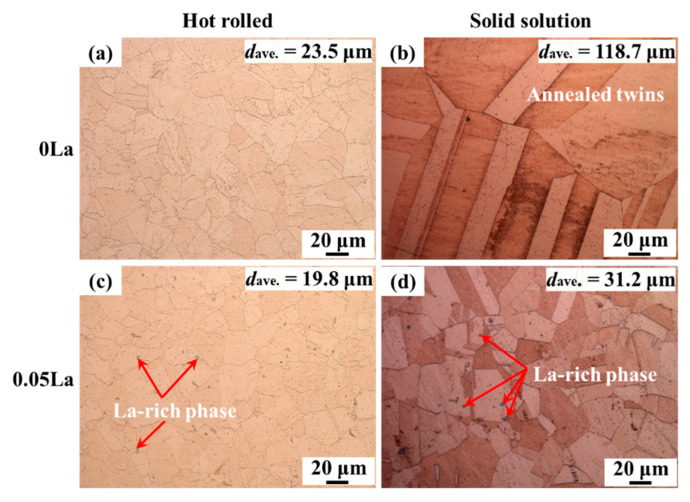
Optical microscope photos of hot-rolled and solid solution alloys: (**a**,**b**) 0 La; (**c**,**d**) 0.05 La.

**Figure 3 materials-16-04105-f003:**
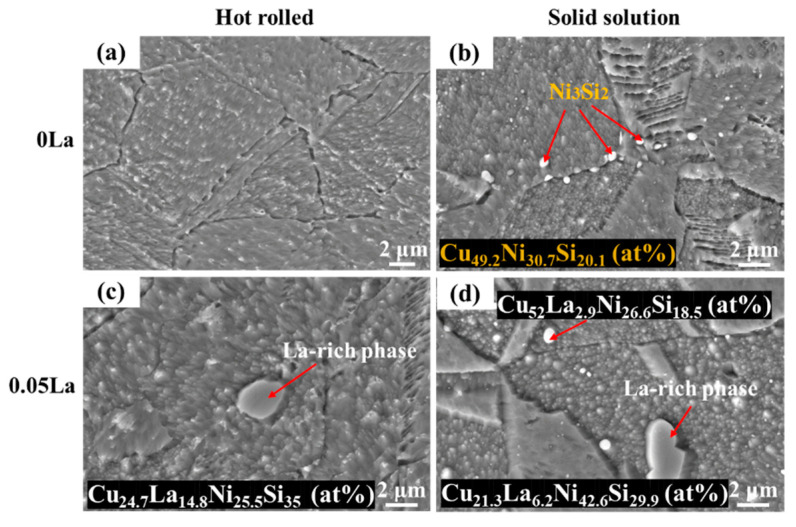
SEM images of hot-rolled and solid solution alloy: (**a**,**b**) 0 La; (**c**,**d**) 0.05 La.

**Figure 4 materials-16-04105-f004:**
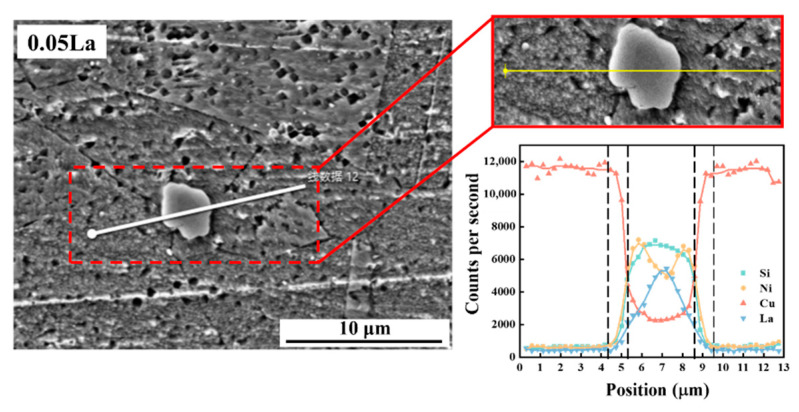
EDS line scanning results of La-rich phase after solid solution treatment.

**Figure 5 materials-16-04105-f005:**
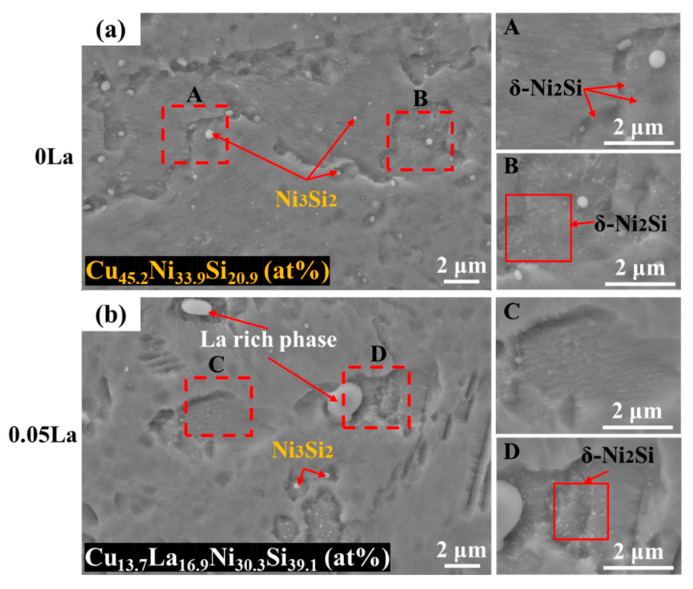
SEM images of aging alloys with different La contents: (**a**) 0 La; (**b**) 0.05 La.

**Figure 6 materials-16-04105-f006:**
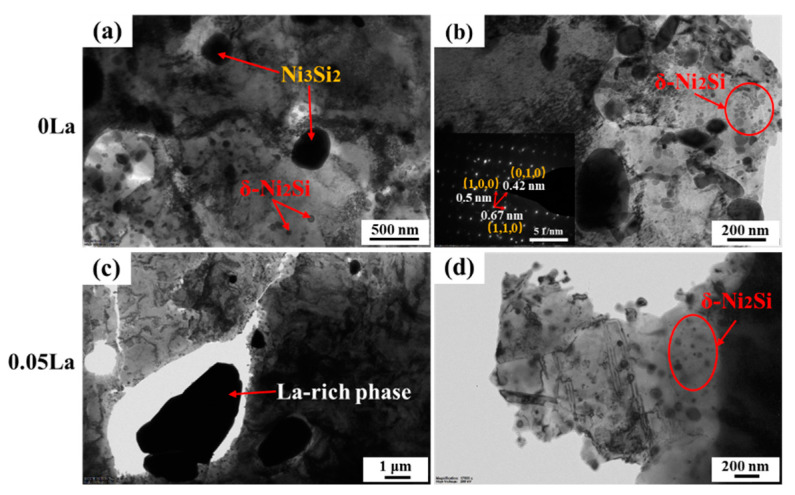
TEM images of the alloys with different La contents after aging treatment: (**a**,**b**) 0 La; (**c**,**d**) 0.05 La.

**Figure 7 materials-16-04105-f007:**
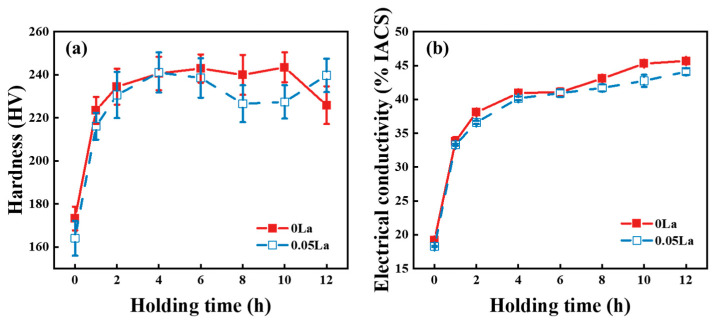
The hardness and electrical conductivity variation curves of alloys with different La contents after aging treatments: (**a**) hardness; (**b**) electrical conductivity.

**Figure 8 materials-16-04105-f008:**
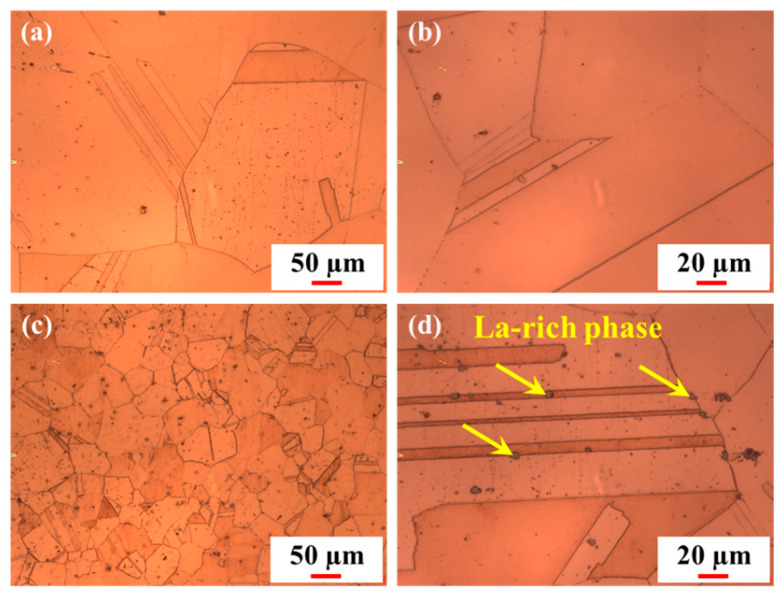
The microstructure of 0 La and 0.05 La alloys after cold rolling and holding at 850 °C for 3 h: (**a**,**c**) 0 La; (**b**,**d**) 0.05 La.

**Figure 9 materials-16-04105-f009:**
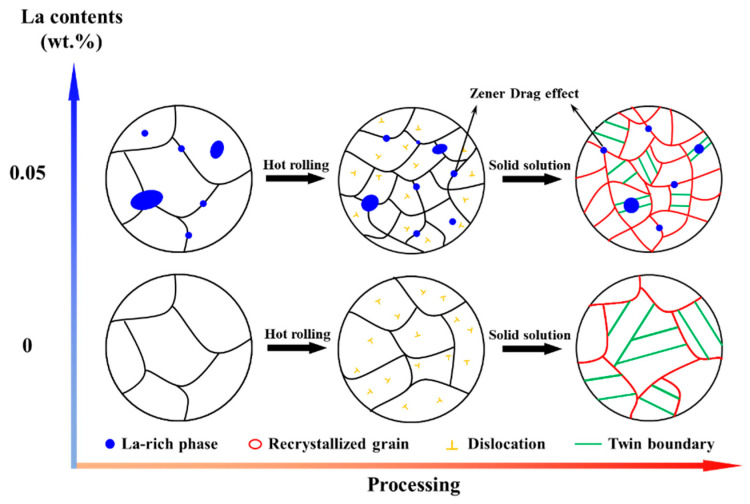
Schematic diagram of La element action during hot rolling and solid solution treatment.

**Figure 10 materials-16-04105-f010:**
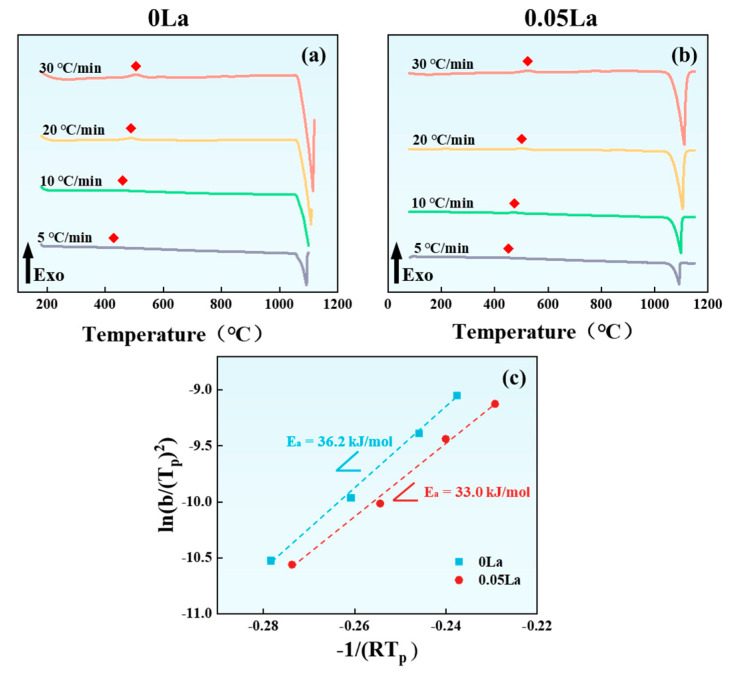
DSC curves: (**a**) 0 La; (**b**) 0.05 La; (**c**) precipitation activation energy of Ni_2_Si phase.

**Figure 11 materials-16-04105-f011:**
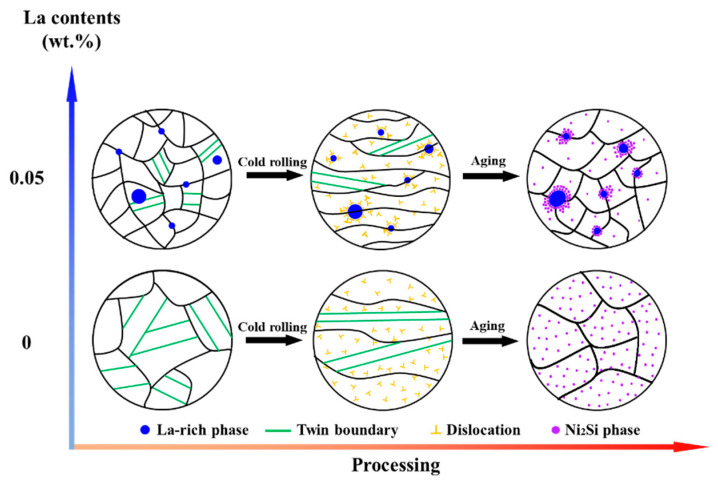
Schematic diagram of La element action during cold rolling and aging treatment.

**Table 1 materials-16-04105-t001:** Specific composition of Cu-Ni-Si alloys with different La contents.

Alloy Type	Ni (wt.%)	Si (wt.%)	Mg (wt.%)	Zn (wt.%)	La (wt.%)	Cu (wt.%)
0 La	2.43	0.56	0.11	0.053	0	Balance
0.05 La	2.48	0.51	0.11	0.048	0.044	Balance

## Data Availability

Not applicable.

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
