# Peer review of "Effects of La Addition on Microstructure Evolution and Thermal Stability of Cu-2.35Ni-0.59Si Sheet"

_materials, 2023, doi:10.3390/ma16114105_

Round 1

Reviewer 1 Report

materials-2367849

 Effects of La addition on microstructure evolution, comprehensive property and thermal stability of Cu-2.35Ni-0.59Si sheet

            In this paper, the authors have worked on a scientific research article: Effects of La addition on microstructure evolution, comprehensive property and thermal stability of Cu-2.35Ni-0.59Si sheet. The manuscript is scientifically proven and analyzed very briefly. The manuscript can be accepted after the authors apply the following changes.

1.               The authors should include the schematic illustration of this scientific article.

2.               Motivation  of this work should be added at the end of the introduction part.

3.               The abstract should be modified and concise according to the finding.

4.               The author should refer to the below paper in introduction section on role of La addition, to strengthen the manuscript. https://doi.org/10.1016/j.physb.2014.04.022, https://doi.org/10.1142/S0217984916502304.

5.               In Section 2.1, the authors need to mention the material manufacturing company/origin

6.               The authors should check the reference style according the MDPI format. Pls check all references.

7.               The authors should metion the EDS result in this sheet material.

8.               The “conclusion” paragraph looks like the summary. It should be brief and give only the final conclusions from the work. If possible add the applications point of view.

9.               Please check the manuscript once again and polish the language. Authors need to give attention to all the spellings in the manuscript carefully.

This manuscript may be accepted after the authors defend the above questions.

 Minor editing of English language required

Author Response

Thank you very much for your suggestions on modifying the article, please see the attachment.

Author Response

(The authors gave the same response as above.)

Reviewer 3 Report

I have read the manuscript titled "Effects of La addition on microstructure evolution, comprehensive property and thermal stability of Cu-2.35Ni-0.59Si sheet " for possible publication journal Materials. I recommend a Major revision for this submission. My comments are as follows.

1. Abstract needs to be modified. It should bring out the summary of the work presented.

2. In keywords, kindly include Cu-Ni-Si as one of the keywords

3. In the introduction, The authors are suggested to discuss the papers on Cu-Ni-Sn and how the work differs from the present system. The authors can discuss the following paper in this manuscript and get some useful papers to discuss Cu-Ni-Sn. The authors in the review article have summarised the Cu-ni-Sn system

Sankar, B., Vinay, C., Vishnu, J., Shankar, K. V., Gokul Krishna, G. P., Govind, V., & Jayakrishna, A. J. (2022). Focused review on Cu–Ni–Sn spinodal alloys: From casting to additive manufacturing. Metals and Materials International, doi:10.1007/s12540-022-01305-6

4. Novelty statement is missing from the introduction section. Please provide it.

5. Why did the authors choose this specific composition?

6. how much load and dwell time did you use for hardness measurement?

7. how did you confirm the phase Ni3Si2 2%. ? did you to XRD? All phases marked in SEM images should be confirmed with XRD

8. did you measure the grain size?

9. For TEM results could you please provide the SDAS pattern 

10. For all SEM images and optical microscopy images, could you please provide the magnification

11. what kind of hardening mechanism is observed in this research?

12. Kindly include error bars for the hardness evaluation

English is satisfactory

Author Response

(The authors gave the same response as above.)

Reviewer 4 Report

The article is full of important content. While I am recommending publication, I think that it need of several major revision that can improve quality and readability of the text: 

(1) It is unclear from the paper of the article what is the reason for the choice of 0.05 wt% La. Why have other concentrations not been investigated?

(2) The methodology is not clear. It is unclear why heat treatment was needed at 850 C when the samples were previously hot rolled and aged? In addition, the article lacks detailed studies of the microstructure (change in phase composition, particle size, volume fraction, etc.) and properties after such heat treatment.

(3) Microstructural studies are weak. The EDS analysis does not provide a complete understanding of the phase composition:

- it is strongly recommended to provide XRD analysis data.

- perhaps the authors should present maps of the distribution of chemical elements, which would clarify the localization of La and other elements in the particles of "La rich phase." Perhaps this is a triple LaxNiySiz connection?

- please give microdifractions and their decoding for Ni3Si2 and "La rich phase." What lattice do "La rich phase" particles have?

(4) In the "Discussion" section, nothing is said about the mechanism of education Ni3Si2. Nothing is noted in the abstract and conclusion about the role of these particles either.

(5) Please specify the measurement error for the data in Figure 7.

Author Response

(The authors gave the same response as above.)

Round 2

Reviewer 1 Report

The authors have addressed all of my concerns with the original manuscript. the revised manuscript is may ready for publication

Author Response

Thank you very much for the valuable modification suggestions provided by the expert. Thank you very much for the expert's recognition of the article.

Reviewer 2 Report

It is my pleasure to accept the manuscript in the current form.

Best Regards

 Minor editing of English language is required.

Author Response

(The authors gave the same response as above.)

Reviewer 3 Report

Dear Authors 

I have read the revised version of the article "Effects of La addition on microstructure evolution, comprehensive property and thermal stability of Cu-2.35Ni-0.59Si sheet" for possible publication in Materials. I appreciate the authors revising the article based on the reviewer's response and accepting that you don't have in-depth knowledge of the SAED pattern. Still, in the SEM and TEM images, the authors have marked certain intermediate phases to prove that these are the phases the authors are requested to provide the EDS analysis for SEM and TEM or the SAED pattern for the TEM. I recently came across a beautiful article https://doi.org/10.1016/j.dt.2023.04.016

Influence of quenching medium on the dendrite morphology, hardness, and tribological behaviour of cast Cu–Ni–Sn spinodal alloy for defence application

In the above article, the authors have clearly explained the SAED pattern of the alloy. I suggest you look at the article's TEM images and SAED pattern and cite and discuss the same. Moreover, I asked the authors to discuss the present alloy with the Cu-Ni-Sn alloys, and I provided you with a review article, but you haven't discussed any Cu-Ni-Sn alloys. Please refer to the review article's reference section and discuss some related papers. There are some beautiful articles related to this system.

Kindly include the images of the samples used for the investigation, it will add more novelty to the work

I can only accept the papers if the above-mentioned comments are addressed. Once again, it is a good article.

English language seems to be fine

Author Response

Thank you very much for the opinions provided by the expert. Please see the attachment.

Reviewer 4 Report

No

Author Response

(The authors gave the same response as above.)

Round 3

Reviewer 3 Report

Accept